# InfoGCL: Information-Aware Graph Contrastive Learning

**Dongkuan Xu**[1]    **Wei Cheng**[2]    **Dongsheng Luo**[1]    **Haifeng Chen**[2]    **Xiang Zhang**[1]

[1]The Pennsylvania State University
[2]NEC Labs America

[1]{dux19,dul262,xzz89}@psu.edu
[2]{weicheng,haifeng}@nec-labs.com

## Abstract

Various graph contrastive learning models have been proposed to improve the performance of learning tasks on graph datasets in recent years. While effective and prevalent, these models are usually carefully customized. In particular, although all recent researches create two contrastive views, they differ greatly in view augmentations, architectures, and objectives. It remains an open question how to build your graph contrastive learning model from scratch for particular graph learning tasks and datasets. In this work, we aim to fill this gap by studying how graph information is transformed and transferred during the contrastive learning process and proposing an information-aware graph contrastive learning framework called InfoGCL. The key point of this framework is to follow the Information Bottleneck principle to reduce the mutual information between contrastive parts while keeping task-relevant information intact at both the levels of the individual module and the entire framework so that the information loss during graph representation learning can be minimized. We show for the first time that all recent graph contrastive learning methods can be unified by our framework. We empirically validate our theoretical analysis on both node and graph classification benchmark datasets, and demonstrate that our algorithm significantly outperforms the state-of-the-arts.

## 1 Introduction

Inspired by their success in the vision and language domains, contrastive learning methods have been wildly adopted by recent progress in graph learning to improve the performance of a variety of tasks [42, 13, 22]. In a nutshell, these methods typically learn representations by creating two augmented views of a graph and maximizing the feature consistency between the two views. Inheriting the advantages of self-supervised learning, contrastive learning relieves graph representation learning from its reliance on label information in graph domain, where label information can be very costly or even impossible to collect while unlabeled/partially labeled data is common, such as chemical graph data [28]. Graph contrastive learning methods have achieved similar (and even better) performance as compared to the equivalent methods trained with labels on benchmark graph datasets [42, 13, 6].

Despite being effective and prevalent, existing graph contrastive learning models differ mostly in augmented view design, encoding architecture, and contrastive objective (refer to Table 1 in Appendix for more comparisons). For a learning task, it usually requires a substantial degree of domain expertise to carefully design and customize these modules for the specific dataset. For example, while both DGI [33] and InfoGraph [28] seek to obtain graph representations by maximizing the mutual information between patch-level and graph-level representations, they adopt different graph encoders, GCN [15] and GIN [38] respectively. mvgrl [13] applies graph diffusion convolution to construct the

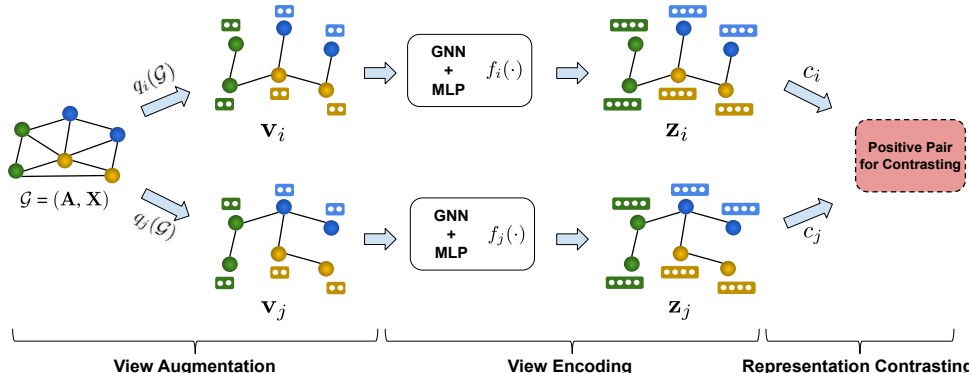

Figure 1: Graph contrastive learning approaches consist of three stages: 1) a graph $\mathcal{G}$ undergoes view augmentation, $q_i(\cdot)$, $q_j(\cdot)$, to obtain two semantically similar views, $\mathbf{v}_i$, $\mathbf{v}_j$. 2) the two views are fed into view encoder networks, $f_i(\cdot)$, $f_j(\cdot)$, to extract latent representations, $\mathbf{z}_i$, $\mathbf{z}_j$. 3) the feature consistency between representations is maximized to optimize the objective function based on contrastive mode $(c_i(\cdot), c_j(\cdot))$, where $c_i(\cdot)$, $c_j(\cdot)$ are aggregation operations applied to representations.

augmented view, while GCC [22] and GRACE [45] adopt subgraph sampling and graph perturbation, respectively.

The main question this paper attempts to answer is: how to perform contrastive learning for your learning tasks on specific graph datasets? However, answering this question is challenging. First, contrastive learning consists of multiple components, such as view augmentation and information encoding. For each of them, there are various choices. Numerous variations make it difficult to design models that are both robust and efficient. Existing graph contrastive learning approaches are carefully designed for different learning tasks on different datasets, however, none of them studies the guiding principles for choosing the best components. Second, graph data has unique properties that distinguish it from other types of data, such as rich structural information and highly diverse distribution [33, 28, 11, 10]. Thus, it is desirable to design the contrastive learning model that fits your graph data properties, even without any domain knowledge of the data.

We propose to address these challenges via Information Bottleneck (IB) [31], which provides a crucial principle for representation learning. Specifically, IB encourages the representations to be maximally informative about the target in the downstream task, which helps keep task-relevant information. Concurrently, IB discourages the representation learning from acquiring the task-irrelevant information from the input data, which is related to the idea of minimal sufficient statistics [27]. However, different from the typical representation learning, there are two information flows involved in the two augmented views in contrastive learning. Therefore, we extend the previous IB work [37, 43] and propose InfoGCL, an information-aware contrastive learning framework for graph data.

To study how information is transformed and transferred, we decouple a typical graph contrastive learning model into three sequential modules (as shown in Figure 1): view augmentation, view encoding, and representation contrasting. We further formalize how to find the optimal of the three modules into three optimization problems. To build the optimal graph contrastive learning model for the particular dataset and task, we argue that it is necessary and sufficient to minimize the mutual information between contrastive representations while maximizing task-relevant information at the levels of both individual module and entire framework. Our work is also motivated by the InfoMin theory [30], which suggests that a good set of views for contrastive learning in the vision domain should share the minimal information necessary to perform well at the downstream task. Beyond view selection, our work extends InfoMin to suggest principles of selecting view encodings and contrastive modes for graph learning considering the unique properties of graph data.

We suggest practically feasible principles to find the optimal modules in graph contrastive learning and show that all recent graph contrastive learning methods can be unified by these principles: i) the augmented views should contain as much task-relevant information as possible, while they should share as little information as possible; ii) the view encoder should be task-relevant and simple as much as possible; iii) the contrastive mode should keep task-relevant information as much as possible after contrasting. Besides, we also investigate the role of negative samples in graph contrastive learning and argue that negative samples are not necessarily required, especially when graph data

is not extremely sparse. Our proposed method, InfoGCL, is validated on a rich set of benchmark datasets for both node-level and graph-level tasks, where we analyze its ability to capture the unique structural properties of the graph data. The results demonstrate that our algorithm achieves highly competitive performance with up to 5.2% relative improvement in accuracy on graph classification task and competitive results on node classification task over the state-of-the-art unsupervised methods.

## 2 Related Work

**Graph Contrastive Learning.** Some recent research efforts in graph domain have been attracted by the success of contrastive learning in vision and language domains [3, 8, 4]. A number of graph contrastive learning approaches have been proposed [28, 22, 42, 13]. Despite all of them creating two views and targeting at maximizing the feature disagreement between the two views, these methods are carefully designed and differ in various aspects. Deep graph Infomax (DGI) [33] applied the InfoMax principle [18] to graph data by contrasting the representations of node-level and graph-level for node classification tasks. Different from DGI, InfoGraph [28] aims at node classification tasks and it contrasts the representations of graph-level and substructure-level of different granularity. In addition, DGI and InfoGraph use different graph encoders to extract latent representations. mvgrl [13] studies both node and graph classification tasks. It transforms the adjacency matrix to a diffusion matrix and treat the two matrices as two congruent views. However, in GCC [22] and GRACE [45], subgraph sampling and graph perturbation are used to create the augmented views. GraphCL [42] explores the view augmentations approaches for graph contrastive learning. Specifically, it studies the approaches of node dropping, edge perturbation, attribute masking, and subgraph sampling. A recent work [44] also studies graph data augmentations. However, it focuses on graph neural networks for node classification and does not study the contrastive learning framework. Our method differs from them. We aim to answer the question how to perform contrastive learning for your graph data and tasks. Instead of carefully designing the architectures, we decouple typical graph contrastive learning into three stages and provide our InfoGCL principles to analyze the optimality theoretically and practically.

**Information Bottleneck.** Our method is related to the Information Bottleneck (IB) theory [31], which aims to find the best trade-off between accuracy and complexity when summarizing a random variable. IB has been recently used to study the deep learning approaches [26, 37, 43]. Specifically, IB expresses the trade-off between the mutual information measures $I(\mathbf{D}, \mathbf{Z})$ and $I(\mathbf{Z}, \mathbf{y})$ as

$$\max \ \mathbf{IB}_\beta = -I(\mathbf{D}; \mathbf{Z}) + \beta I(\mathbf{Z}; \mathbf{y}), \tag{1}$$

where $\mathbf{D}$, $\mathbf{Z}$, $\mathbf{y}$ are the input, the latent representation and the task label, respectively. $\theta$ is a hyperparameter. In other words, IB aims to learn representation $\mathbf{Z}$ that is maximally expressive about $\mathbf{y}$, while being minimally expressive about $\mathbf{D}$. More recently, there are some efforts applying the IB theory to graph representation learning. [37] aims to generate both expressive and robust graph representations, while [43] studies the subgraph recognition problem. In contrast, our work focuses on graph contrastive learning, where the two augmented views make the optimization objective (information trade-off) different. Our work is also related to the idea of minimal sufficient statistics [27], which has been recently studied in the vision domain [30], claiming that a good set of image views should share the minimal information necessary to perform well at the downstream task. Different from [30], we focus on the graph domain and propose three stages considering the unique properties of graph data.

## 3 Preliminaries and Notations

**Graph Representation Learning.** A graph is denoted by $\mathcal{G} = (\mathbf{A}, \mathbf{X})$. $\mathbf{A} \in \mathbb{R}^{n \times n}$ is the adjacency matrix. $\mathbf{X} \in \mathbb{R}^{n \times d}$ is the node attribute matrix, where $d$ is the attribute dimension. In this work, we focus on both node-level and graph-level tasks. For node-level task, given graph $\mathcal{G}$ and the labels of a subset of nodes, denoted by $\mathbf{Y}_v$, the goal is to learn the latent representation $\mathbf{z}_v$ for each node $v$ such that $\mathbf{z}_v$ preserves both network structures and node attributes, which can be further used to predict $\mathbf{Y}_v$. For graph-level task, given a set of graphs $\mathbb{G} = \{\mathcal{G}^1, \mathcal{G}^2, \cdots \}$ and the labels of some graphs, denoted by $\mathbf{Y}_g$, the goal is to learn the latent representation $\mathbf{z}_g$ for each graph such that $\mathbf{z}_g$ can be used to predict $\mathbf{Y}_g$. Typically, the graph data is fed into graph neural networks (GNNs) to generate the representations, such as $\mathbf{z}_g = \text{GNNs}(\mathcal{G})$.

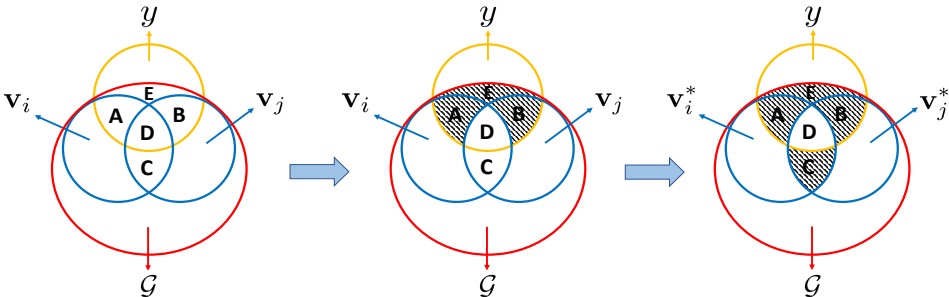

Figure 2: Illustration of optimal views. (*Left*) The relationships between graph $\mathcal{G}$, two views, $\mathbf{v}_i$, $\mathbf{v}_j$, and task $y$ in terms of information entropy. A, B, C, D, E are overlapping areas. Two views are contained by graph because views are functions of graph. (*Middle*) A, B, E become null when Eqs. (4)-(5) hold, which indicates the views and graph share the same amount of task-relevant information. (*Right*) C further becomes null when Eq. (3) holds, which indicates all the shared information between views is task-relevant, i.e., the views are optimal.

**Graph Contrastive Learning.** Given an input graph, graph contrastive learning aims to learn the representations of graph or nodes (for graph-level or node-level tasks respectively) through maximizing the feature consistency between two augmented views of the input graph via contrastive loss in the latent space. We decouple a typical graph contrastive learning model into three sequential modules.

**(i) View augmentation**. Graph $\mathcal{G}$ undergoes data augmentation $q(\cdot)$ to obtain two views $\mathbf{v}_i$, $\mathbf{v}_j$, i.e., $\mathbf{v}_i \sim q_i(\mathcal{G})$ and $\mathbf{v}_j \sim q_j(\mathcal{G})$. A view is represented as graph data, such as $\mathbf{v}_i = (\mathbf{A}_{v_i}, \mathbf{X}_{v_i})$, where $\mathbf{A}_{v_i} \in \mathbb{R}^{n \times n}$ and $\mathbf{X}_{v_i} \in \mathbb{R}^{n \times d}$. In practice, view augmentation approaches include node dropping, edge perturbation, subgraph sampling, etc.

**(ii) View encoding**. Graph-level or node-level latent representation is extracted from views $\mathbf{v}_i$, $\mathbf{v}_j$ by using the view encoder networks $f(\cdot)$ (a GNN backbone plus a projection MLP), i.e., $\mathbf{z}_i \sim f_i(\mathbf{v}_i)$ and $\mathbf{z}_j \sim f_j(\mathbf{v}_j)$. The two encoders might or might not share parameters depending on whether they are from the same domain.

**(iii) Representation contrasting**. Given the latent representations, a contrastive loss is optimized to score the positive pairs $\mathbf{z}_i$, $\mathbf{z}_j$ higher compared to other negative pairs. Typically, the negative pairs are constructed from the augmented views of other graphs in the same minibatch. The InfoNCE loss [21] has been adopted as one of popular contrastive losses, which is defined as:

$$\mathcal{L}_{NCE} = -\mathbb{E}\left[log\frac{exp(h(\mathbf{z}_{i,n}, \mathbf{z}_{j,n}))}{\sum_{n'=1}^{N} exp(h(\mathbf{z}_{i,n}, \mathbf{z}_{j,n'}))}\right], \tag{2}$$

where $h(\cdot)$ is a contrasting operation to score the agreement between two representations. Theoretically, minimizing the InfoNCE loss equivalently maximizes a lower bound on the mutual information between the views of positive pairs. In other words, $I(\mathbf{z}_i, \mathbf{z}_j) \geqslant log(N) - \mathcal{L}_{NCE}$, where $I(\cdot)$ measures the mutual information.

# 4 Information-Aware Graph Contrastive Learning

In this paper, we study how to perform contrastive learning for specific graph tasks and datasets. In particular, we attempt to answer the following questions for graph contrastive learning: (i) What is the optimal augmented views? (ii) What is the optimal view encoder? (iii) What is the optimal contrastive mode?

## 4.1 View Augmentation

The goal of view augmentation is to create realistically rational data via the transformation approaches that do not affect the semantic label. Compared to the augmentation in other domains, graph view augmentation needs to consider the structural information of graph data, such as the node, the edge, and the subgraph. There are various graph view augmentation methods proposed recently. We follow

a similar definition used in [42] to categorize four kinds of view augmentation approaches for graph data. **Node dropping** discards a certain part of nodes along with their edges in the input graph to create a new graph view. **Edge perturbation** perturbs the connectivity in the graph via adding or dropping partial edges. **Attribute masking** masks part of node attributes and assumes that the missing attributes can be well predicted by the remaining ones. **Subgraph sampling** samples a subgraph from the input graph. The rationale behind these approaches is that the semantic meaning of graph has certain robustness to graph perturbation.

The augmented views generated in the graph contrastive framework are typically used in a separate downstream task. To characterize what views are optimal for a downstream task, we define the optimality of views. The main motivation is: the optimal augmented views should contain the most task-relevant information, and the information shared between views should only be task-relevant.

**Corollary 1.** *(Optimal Augmented Views) For a downstream task $T$ whose goal is to predict a semantic label $y$, the optimal views, $\mathbf{v}_i^*$, $\mathbf{v}_j^*$, generated from the input graph $\mathcal{G}$ are the solutions to the following optimization problem :*

$$(\mathbf{v}_i^*, \mathbf{v}_j^*) = \arg\min_{\mathbf{v}_i, \mathbf{v}_j} I(\mathbf{v}_i; \mathbf{v}_j) \tag{3}$$

$$s.t. \ \ I(\mathbf{v}_i; y) = I(\mathbf{v}_j; y) \tag{4}$$

$$I(\mathbf{v}_i; y) = I(\mathcal{G}; y) \tag{5}$$

This says that for the optimal graph views, the amount of information shared between them is minimized (Eq. (3)), while the two views contain the same amount of information with respect to $y$ (Eq. (4)), which is also the amount of information that the input gprah contains about the task (Eq. (5)). The illustration of the optimal views is shown in Figure 2 and the proof is in the Appendix.

## 4.2 View Encoding

View encoding aims to extract the latent representations of nodes or graphs via feeding the data of two views into view encoder networks such that the generated representations preserve both structure and attribute information in the views. The view encoders are quite flexible in graph contrastive learning and typically they are GCN [15], GAT [32], or GIN [38], etc.

The representations extracted via view encoding are further utilized to optimize the objective function of contrastive learning. After well trained, the view encoders are used to generate the graph/node representations for a downstream task. To characterize what encoders are optimal, we define the optimality of view encoders for graph contrastive learning. The main motivation is: the representation generated by the optimal encoder for a view should keep all the shared information by the two contrastive views, meanwhile the kept information is all task-relevant.

**Corollary 2.** *(Optimal View Encoder) Given the optimal views, $\mathbf{v}_i^*$, $\mathbf{v}_j^*$, for a downstream task $T$ whose goal is to predict a semantic label $y$, the optimal view encoder for view $\mathbf{v}_i^*$ is the solution to the following optimization problem :*

$$f_i^* = \arg\min_{f_i} I(f_i(\mathbf{v}_i^*); \mathbf{v}_i^*) \tag{6}$$

$$s.t. \ \ I(f_i(\mathbf{v}_i^*); \mathbf{v}_j^*) = I(\mathbf{v}_i^*; \mathbf{v}_j^*) \tag{7}$$

It indicates that for the optimal view encoder, the amount of information shared between the optimal view and the extracted representation is minimized (Eq. (6)), while the information shared between the two optimal views is kept after the encoding process of one view (Eq. (7)). The illustration of the optimal encoder is shown in Figure 3 and the proof is illustrated in the Appendix.

## 4.3 Representation Contrasting

To allow flexible contrasting for graph data, we consider contrastive modes similar to [13]. A contrastive mode is denoted by $(c_i(\cdot), c_j(\cdot))$, where $c_i(\cdot), c_j(\cdot)$ are the aggregation operations applied to the representations extracted by view encoders, The contrastive modes are unique to graph data because of the structural information inside a graph. Specifically, we consider five contrastive modes. In **global-global mode**, the graph representations from two views are contrasted. Thus, $c_i(\cdot), c_j(\cdot)$ are averaging aggregation operations in this mode. In **local-global mode**, we contrast the node

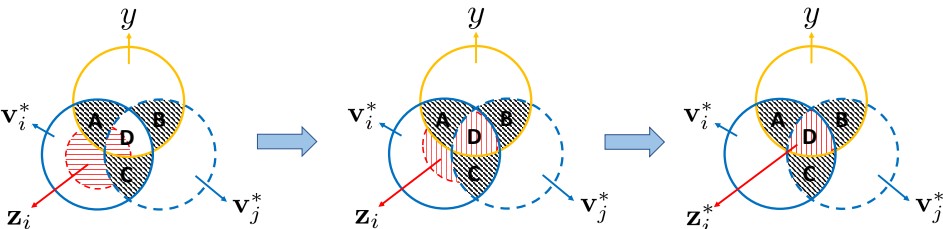

Figure 3: Illustration of optimal view encoding. (*Left*) The relationships between two (optimal) views, $\mathbf{v}_i^*$, $\mathbf{v}_j^*$, task $y$, and representation $\mathbf{z}_i$ in terms of information entropy. A, B, C are null because the two views are optimal here. $\mathbf{z}_i$ is contained by view $\mathbf{v}_i^*$ because representations are functions of views. (*Middle*) $\mathbf{z}_i$ covers D when Eq. (7) holds, which indicates the shared information between views is kept after encoding. (*Right*) View $\mathbf{z}_i$ further exactly covers D and the view encoding becomes optimal, i.e., $\mathbf{z}_i^*$, which indicates all the information shared between view $\mathbf{v}_i^*$ and representation $\mathbf{z}_i^*$ is task-relevant.

representations from one view with the graph representations from the other view. Thus, $c_i(\cdot)$, $c_j(\cdot)$ are the identical transformation and averaging aggregation operations, respectively. In **local-local mode**, the node representations from two views are contrasted. In **multi-scale mode**, we contrast graph representation of one view with the intermediate representation from the other. In **hybrid mode**, both global-global and local-global are applied.

To characterize which mode is optimal, we define the optimality of contrastive mode for graph contrastive learning. The main motivation is: the optimal contrastive mode keeps the most task-relevant information after the representations are aggregated. The proof is included in the Appendix.

**Corollary 3.** (*Optimal Contrastive Mode*) *Given the latent representations, $\mathbf{z}_i^*$, $\mathbf{z}_j^*$, extracted by the optimal view encoders, i.e., $\mathbf{z}_i^* = f_i^*(\mathbf{v}_i^*)$, $\mathbf{z}_j^* = f_j^*(\mathbf{v}_j^*)$, and a downstream task $T$ with label $y$, the optimal contrastive mode is the solution to the following optimization problem, where $c_i$, $c_j$ are the aggregation operations applied to the latent representations:*

$$(c_i^*, c_j^*) = \underset{(c_i, c_j)}{\arg\min} -I(c_i(\mathbf{z}_i^*); c_j(\mathbf{z}_j^*)). \tag{8}$$

### 4.4 InfoGCL Principle

According to our proposed corollaries, we can theoretically design the optimal contrastive learning approach for our specific graph data and task. However, in real-world scenarios, the conditions to meet the exact optimality of contrastive learning is hard or even not practically possible to reach because of data noise and limited model capability. Therefore, we propose to achieve the optimal for each stage independently and practically, which is an approximation to achieve the original optimality. Specifically, we make the following propositions to address the questions of the optimal views, optimal view encoder, and optimal contrastive mode.

**Proposition 1.** *For a task $T$ with label $y$, given a bunch of graph view augmentation methods, $\{q_1(\cdot), q_2(\cdot), \cdots \}$, that create two views $\mathbf{v}_i$, $\mathbf{v}_j$, the recommended augmentation methods are the ones, $q_i(\cdot)$, $q_j(\cdot)$, that maximize $I(\mathbf{v}_i; y) + I(\mathbf{v}_j; y) - I(\mathbf{v}_i; \mathbf{v}_j)$, i.e., the area of $A+B+D$ in Figure 2.*

**Proposition 2.** *Given a task $T$ with label $y$ and a set of view encoders, $\{f_i^1(\cdot), f_i^2(\cdot), \cdots \}$, that generate representation $\mathbf{z}_i$ via taking view $\mathbf{v}_i$ as input, the recommended view encoder is the one that maximizes the mutual information between $\mathbf{v}_i$, $\mathbf{z}_i$ and $y$. Symmetrically the same for view $\mathbf{v}_j$.*

**Proposition 3.** *Given a task $T$ with label $y$, the extracted representations, $z_i$, $z_j$, and a set of aggregation operations, $\{c_1(\cdot), c_2(\cdot), \cdots \}$, the recommended contrastive mode is the one, $(c_i, c_j)$, that has the largest amount of mutual information between $c_i(\mathbf{z}_i)$, $c_j(\mathbf{z}_j)$ and $y$.*

The qualitative and quantitative evaluation of these propositions are shown in Section 5.3.

### 4.5 Role of Negative Samples

Current graph contrastive learning approaches heavily depend on negative samples. However, recent progresses of contrastive learning in vision domain indicate that negative samples are not necessarily

|  | Graph Task Datasets | | | | | | Node Task Datasets | | |
|---|---|---|---|---|---|---|---|---|---|
|  | MUTAG | PTC-MR | IMDB-B | IMDB-M | NCI1 | COLLAB | Cora | Citeseer | Pubmed |
| # Graphs | 188 | 344 | 1000 | 1500 | 4110 | 5000 | 1 | 1 | 1 |
| # Nodes | 17.9 | 14.3 | 19.8 | 13.0 | 29.9 | 74.5 | 3327 | 2708 | 19717 |
| # Edges | 19.8 | 14.7 | 193.1 | 65.9 | 1.1 | 33.0 | 4732 | 5429 | 44338 |
| # Classes | 2 | 2 | 2 | 2 | 2 | 3 | 6 | 7 | 3 |

Table 1: Dataset statistics.

required [8, 4], of which the main benefit is to avoid careful treatment to retrieve the negative pairs. To study the influence of negative samples on graph contrastive learning, we follow the framework of SimSiam [4] to revise the loss function as Eq.(9). A very recent work [29] also studies graph contrastive learning without negative samples. Different from it, we focus on both node and graph classification tasks.

$$\mathcal{L} = -\frac{1}{N} \sum_{n=1}^{N} \frac{\mathbf{z}_{i,n}}{\|\mathbf{z}_{i,n}\|} \cdot \frac{\mathbf{z}_{j,n}}{\|\mathbf{z}_{j,n}\|}, \tag{9}$$

# 5 Experiments

In this section, we evaluate our InfoGCL with a number of experiments. We first describe datasets, evaluation protocol, and experimental setup. Then, we present the experimental results on both node and graph classification. Last, we analyze our proposed principles via ablation study.

## 5.1 Setup

We use both graph classification and node classification benchmark datasets that are widely used in the existing graph contrastive learning approaches. The graph classification datasets include MUTAG [17], PTC-MR [17], IMDB-B [40], IMDB-M [40], NCI1 [34], and COLLAB [40]. MUTAG is a collection of nitroaromatic compounds represented as graphs, where vertices stand for atoms and edges represent bonds between atoms. PTC-MR is a collection of 344 chemical compounds which report the carcinogenicity for rats. IMDB-B and IMDB-M are two movie collaboration datasets, where nodes represent actors/actress and there is an edge between them if they appear in the same movie. In NCI1, graphs are the representation of chemical compounds, where vertices stand for atoms and edges represent bonds between atoms. COLLAB is a collaboration dataset, where researchers are nodes and an edge indicates collaboration between two researchers. The node classification datasets include Citeseer, Cora, and Pubmed [23]. All of them are citation networks, where nodes are documents and edges are citation links. These datasets are summarized in Table 1.

We closely follow the evaluation protocol of previous state-of-the-art graph contrastive learning approaches. For graph classification, we report the mean 10-fold cross validation accuracy after 5 runs followed by a linear SVM. The linear SVM is trained by applying cross validation on training data folds and the best mean accuracy is reported. For node classification, we report the mean accuracy on test set after 50 runs of training followed by a linear neural network model. To make comparison fair, we adopt the basic setting of InfoGraph for graph classification. We conduct experiment with the values of the number of GNN layers, the number of epochs, batch size, the parameter C of SVM in the sets $\{2, 4, 8, 12\}$, $\{10, 20, 40, 100\}$, $\{32, 64, 128, 256\}$ and $\{10^{-3}, 10^{-2}, ..., 10^2, 10^3\}$, respectively. We adopt the basic setting of DGI for node classification. Specifically, we set the number of GNN layers to 1 and experiment with the batch size in the set $\{2, 4, 8\}$. The hidden dimension of representations is set to 512. We also apply the early stopping strategy.

## 5.2 Experimental Results

To evaluate our method InfoGCL on graph classification, we use thhree categories of baselines. The kernel approaches include shortest path kernel (SP) [2], Graphlet kernel (GK) [25], Weisfeiler-Lehman sub-tree kernel (WL) [24], deep graph kernels (DGK) [39], and multi-scale Laplacian kernel (MLG) [16]. The supervised baselines include GraphSAGE [12], GCN [15], GIN [38], GAT [32]. We also compare with the unsupervised approaches, including RandomWalk [7], node2vec [9], sub2vec [14], graph2vec [20], InfoGraph [28], GraphCL [42], and mvgrl [13]. Table 2 shows the

| Method | MUTAG | PTC-MR | IMDB-B | IMDB-M | NCI1 | COLLAB |
|---|---|---|---|---|---|---|
| **Kernel Approaches** | | | | | | |
| SP | $85.2 \pm 2.4$ | $58.2 \pm 2.4$ | $55.6 \pm 0.2$ | $38.0 \pm 0.3$ | $73.5 \pm 0.1$ | - |
| GK | $81.7 \pm 2.1$ | $57.3 \pm 1.4$ | $65.9 \pm 1.0$ | $43.9 \pm 0.4$ | $66.0 \pm 0.1$ | $72.8 \pm 0.3$ |
| WL | $80.7 \pm 3.0$ | $58.0 \pm 0.5$ | $\mathbf{72.3 \pm 3.4}$ | $\mathbf{47.0 \pm 0.5}$ | $80.0 \pm 0.5$ | $\mathbf{78.9 \pm 1.9}$ |
| DGK | $87.4 \pm 2.7$ | $60.1 \pm 2.6$ | $67.0 \pm 0.6$ | $44.6 \pm 0.5$ | $80.3 \pm 0.5$ | $73.1 \pm 0.3$ |
| MLG | $\mathbf{87.9 \pm 1.6}$ | $\mathbf{63.3 \pm 1.5}$ | $66.6 \pm 0.3$ | $41.2 \pm 0.0$ | $\mathbf{80.8 \pm 1.3}$ | - |
| **Supervised Approaches** | | | | | | |
| GraphSAGE | $85.1 \pm 7.6$ | $63.9 \pm 7.7$ | $72.3 \pm 5.3$ | $50.9 \pm 2.2$ | $77.7 \pm 1.5$ | $68.3 \pm 4.2$ |
| GCN | $85.6 \pm 5.8$ | $64.2 \pm 4.3$ | $74.0 \pm 3.4$ | $51.9 \pm 3.8$ | $80.2 \pm 2.0$ | $79.0 \pm 1.8$ |
| GIN-0 | $\mathbf{89.4 \pm 5.6}$ | $64.6 \pm 7.0$ | $\mathbf{75.1 \pm 5.1}$ | $\mathbf{52.3 \pm 2.8}$ | $\mathbf{82.7 \pm 1.7}$ | $\mathbf{80.2 \pm 1.9}$ |
| GIN-e | $89.0 \pm 6.0$ | $63.7 \pm 8.2$ | $74.3 \pm 5.1$ | $52.1 \pm 3.6$ | $\mathbf{82.7 \pm 1.6}$ | $80.1 \pm 1.9$ |
| GAT | $\mathbf{89.4 \pm 6.1}$ | $\mathbf{66.7 \pm 5.1}$ | $70.5 \pm 2.3$ | $47.8 \pm 3.1$ | $66.6 \pm 2.2$ | $67.4 \pm 2.9$ |
| **Unsupervised Approaches** | | | | | | |
| RandomWalk | $83.7 \pm 1.5$ | $57.9 \pm 1.3$ | $50.7 \pm 0.3$ | $34.7 \pm 0.2$ | $64.3 \pm 0.3$ | - |
| node2vec | $72.6 \pm 10.2$ | $58.6 \pm 8.0$ | $50.2 \pm 0.9$ | $36.0 \pm 0.7$ | $54.9 \pm 1.6$ | $56.1 \pm 0.2$ |
| sub2vec | $61.1 \pm 15.8$ | $60.0 \pm 6.4$ | $55.3 \pm 1.5$ | $36.7 \pm 0.8$ | $52.8 \pm 1.5$ | - |
| graph2vec | $83.2 \pm 9.6$ | $60.2 \pm 6.9$ | $71.1 \pm 0.5$ | $50.4 \pm 0.9$ | $75.4 \pm 1.2$ | - |
| InfoGraph | $89.0 \pm 1.1$ | $61.7 \pm 1.4$ | $73.0 \pm 0.9$ | $49.7 \pm 0.5$ | $76.2 \pm 1.4$ | $70.7 \pm 1.1$ |
| GraphCL | $86.8 \pm 1.3$ | $61.3 \pm 2.1$ | $71.1 \pm 0.4$ | $49.2 \pm 0.6$ | $77.9 \pm 0.4$ | $71.4 \pm 1.2$ |
| mvgrl | $89.7 \pm 1.1$ | $62.5 \pm 1.7$ | $74.2 \pm 0.7$ | $51.2 \pm 0.5$ | $77.0 \pm 0.8$ | $76.0 \pm 1.2$ |
| InfoGCL | $\mathbf{91.2 \pm 1.3}$ | $\mathbf{63.5 \pm 1.5}$ | $\mathbf{75.1 \pm 0.9}$ | $\mathbf{51.4 \pm 0.8}$ | $\mathbf{80.2 \pm 0.6}$ | $\mathbf{80.0 \pm 1.3}$ |

Table 2: Graph classification results (%).

| Method | Cora | Citeseer | Pubmed |
|---|---|---|---|
| **Supervised Approaches** | | | |
| MLP | 55.1 | 46.5 | 71.4 |
| ICA | 75.1 | 69.1 | 73.9 |
| LP | 68.0 | 45.3 | 63.0 |
| ManiReg | 59.5 | 60.1 | 70.7 |
| SemiEmb | 59.0 | 59.6 | 71.7 |
| Planetoid | 75.7 | 64.7 | 77.2 |
| Chebyshev | 81.2 | 69.8 | 74.4 |
| GCN | 81.5 | 70.3 | $\mathbf{79.0}$ |
| JKNet | $82.7 \pm 0.4$ | $\mathbf{73.0 \pm 0.5}$ | $77.9 \pm 0.4$ |
| GAT | $\mathbf{83.0 \pm 0.7}$ | $72.5 \pm 0.7$ | $\mathbf{79.0 \pm 0.3}$ |
| **Unsupervised Approaches** | | | |
| Linear | $47.9 \pm 0.4$ | $49.3 \pm 0.2$ | $69.1 \pm 0.3$ |
| DeepWalk | $70.7 \pm 0.6$ | $51.4 \pm 0.5$ | $74.3 \pm 0.9$ |
| GAE | $71.5 \pm 0.4$ | $65.8 \pm 0.4$ | $72.1 \pm 0.5$ |
| VERSE | $72.5 \pm 0.3$ | $55.5 \pm 0.4$ | - |
| DGI | $83.8 \pm 0.5$ | $72.0 \pm 0.6$ | $77.9 \pm 0.3$ |
| GraphCL | $82.5 \pm 0.1$ | $73.1 \pm 0.2$ | - |
| mvgrl | $\mathbf{86.8 \pm 0.5}$ | $73.3 \pm 0.5$ | $\mathbf{80.1 \pm 0.7}$ |
| InfoGCL | $83.5 \pm 0.3$ | $\mathbf{73.5 \pm 0.4}$ | $79.1 \pm 0.2$ |

Table 3: Node classification results (%).

graph classification results. We observe that our approach achieves the best results compared to other unsupervised approaches. Our approach also outperforms or matches the best kernel approaches across the datasets. Even compared with the supervised ones, our approach achieves the best in 2 out of 6 datasets and the results of our approach on other 4 dataset are among the top.

For node classification tasks, we compare InfoGCL with some supervised approaches and unsupervised approaches. The supervised baselines include a simple MLP model, iterative classification algorithm (ICA) [19], manifold regularization (ManiReg) [1], semi-supervised embedding (SemiEmb) [35], Planetoid [41], Chebyshev [5], GCN, JKNet [36], GAT. Table 3 shows the node classification results. It is observed that our approach achieves the state-of-the-art results and competes the best one with respect to the existing unsupervised approaches. Compared to supervised baselines, our approach outperforms all the baselines.

| Method | MUTAG | IMDB-B | COLLAB | Cora | Citeseer | Pubmed |
|---|---|---|---|---|---|---|
| InfoGCL (w/o neg) | $91.0 \pm 1.4$ | $75.1 \pm 0.5$ | $80.2 \pm 1.0$ | $78.6 \pm 0.4$ | $70.4 \pm 0.6$ | $77.4 \pm 0.7$ |
| InfoGCL (w/ neg) | $91.2 \pm 1.3$ | $75.1 \pm 0.9$ | $80.0 \pm 1.3$ | $83.5 \pm 0.3$ | $73.5 \pm 0.4$ | $79.1 \pm 0.2$ |

Table 4: Comparison between InfoGCL with negative samples and without negative samples.

## 5.3 Evaluation of InfoGCL Principle

We can unify the existing graph contrastive learning methods through the perspective of InfoGCL principle: all recent graph contrastive learning methods can be decoupled into three stages that implicitly follow the InfoGCL principle, though being different in model architecture design and optimization strategies. Below, we analyze some observations from several recent work. Because of the limited space, please refer to the Appendix for more results of the quantitative analysis.

**Obs. i.** Composing a graph and its augmentation benefits downstream performance [42]. Compared to composing a graph and the graph itself, augmentation leads to smaller $I(\mathbf{v}_i; \mathbf{v}_j)$ (Proposition 1).

**Obs. ii.** Composing different augmentations benefits more [42]. Compared to composing a graph and its augmentations, two augmentations further decrease $I(\mathbf{v}_i; \mathbf{v}_j)$ (Proposition 1).

**Obs. iii.** Node dropping and subgraph sampling are generally beneficial across datasets [42]. When compared to attribute masking and edge perturbation, they change the semantic meaning of the graph relatively slightly, which leads to higher $I(\mathbf{v}_i; y)$, $I(\mathbf{v}_j; y)$ (Proposition 1).

**Obs. iv.** Edge perturbation benefits social networks but hurts some biochemical molecules [42]. The semantic meaning of social networks are robust to edge perturbation. However, the semantic meaning of some biochemical molecules are determined by local connection pattern, where edge perturbation decreases $I(\mathbf{v}_i; y)$ (Proposition 1).

**Obs. v.** Contrasting node and graph representations consistently performs better than other contrastive modes across benchmarks [13]. Compared to other contrastive modes, node-graph (i.e., local-global) mode generally extracts more graph structure information, which benefits predicting task label $y$ (Proposition 3).

## 5.4 Effect of Negative Samples

To study the effect of negative samples on graph contrastive learning, we follow SimSiam [4] and design the objective as Eq. (9). We conduct experiments on three graph task and three node task datasets. The results are reported in Table 4. It is observed that the negative samples show little influence on the three graph task datasets, while performance drops on the three node task datasets, especially the Cora dataset.

According to the dataset statistics summarized in Table 1, we see the networks of Cora, Citeseer, Pubmed are much sparser (in terms of network topology). Furthermore, we know the node features of these three datasets are also much sparser (one-hot encoding with high dimensionality). We speculate this because the contrastive learning models tends to collapse easier if negative samples are not used, especially when data is too sparse. Therefore,we make the hypothesis: negative samples benefit graph modeling, especially when i) network topology, and ii) node features are extremely sparse.

## 6 Conclusion and Limitations

We propose InfoGCL, an information-aware graph contrastive learning framework for graph contrastive learning. Existing graph contrastive learning approaches are usually carefully designed. We aim to answer how to perform contrastive learning for your learning tasks on specific graph data. Our method decouples the typical contrastive learning approaches into three sequential modules and provides the theoretical analysis for reaching the optimality. To address the questions of optimality in a practical way, we propose the InfoGCL principle, which is implicitly followed by all recent graph contrastive learning approaches. In addition, we explore the role of negative samples in graph contrastive learning and find negative samples are not necessarily required. Experiments on both node and graph benchmark datasets demonstrate the effectiveness of our method. Note that our method is not without limitations. We can further improve our method by designing better practical approximations to the theoretical optimality of graph contrastive learning.

## Funding Transparency Statement

This project was partially supported by NSF projects IIS-1707548 and CBET-1638320.

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
