# Appendix: Information-Aware Graph Contrastive Learning

**Dongkuan Xu**[1]   **Wei Cheng**[2]   **Dongsheng Luo**[1]   **Haifeng Chen**[2]   **Xiang Zhang**[1]

[1]The Pennsylvania State University
[2]NEC Labs America
[1]{dux19,dul262,xzz89}@psu.edu
[2]{weicheng,haifeng}@nec-labs.com

## 1   Comparison of Graph Contrastive Learning Approaches

We summarize the published approaches of graph contrastive learning in Table 1. Specifically, the tasks include graph-level and node-level tasks. The GNNs are used to encode the graph data and they are quite flexible. A typical augmented view is the graph itself, but it focuses on different structure-level of the graph. The two views adopted by mvgrl are the graph itself and the diffused graph, respectively. The four views used by GraphCL are discussed in the text. GCA applies edges removing and attribute masking to generate two views. In CSSLs, edge deletion/insertion and node deletion/insertion are used to generate different views. Similar to GCA, SelfTask also applies edges removing and attribute masking to generate views. The contrastive modes are discussed in the text, which include global-global, local-global, local-local, multi-scale mode, and hybrid modes. mvgrl studies the effect of various contrastive modes. SelfTask uses both local-local and global-global. The additional comparison of sampling strategies and objective functions between published graph contrastive learning approaches is shown in Table-2.

| Approach | Task | GNN | Augmented Views | Contrastive Mode |
|---|---|---|---|---|
| DGI [6] | Node | GCN | Graph itself | Global-local |
| InfoGraph [4] | Both | GIN | Graph itself | Multi-scale |
| mvgrl [1] | Both | GCN | Two views | Various |
| GCC [3] | Both | GIN | Graph itself | Local-local |
| GRACE [9] | Node | GCN, GraphSAGE | Graph itself | Local-local |
| GraphCL [7] | Both | GCN, GIN, GAT | Four views | Global-global |
| GCA [10] | Node | GCN | Two views | Local-local |
| CSSLs [8] | Graph | HGP-SL | Four views | Global-global |
| SelfTask [2] | Node | GCN | Two views | Two modes |

Table 1: A comparison of published approaches for graph contrastive learning.

| Approach | Sampling Strategy | Obj. Function |
|---|---|---|
| DGI [6] | Randomly sampled graphs (or a graph transformation) | JSD |
| InfoGraph [4] | Global and local patch across all graph instances in a batch | JSD |
| mvgrl [1] | Joint distribution for positive, product of marginals for negative | JSD, InfoNCE |
| GCC [3] | Randomly sampled graphs | InfoNCE |
| GRACE [9] | Negative samples are all other nodes in the two views | NT-xent |
| GraphCL [7] | N-1 augmented graphs within the same minibatch | NT-xent |
| GCA [10] | Negative samples are all other nodes in the two views | InfoNCE |
| CSSLs [8] | Randomly sampled graphs | NT-xent |
| SelfTask [2] | Randomly sampled edges/attributes | Cross Entropy |

Table 2: Different sampling strategies and objective functions of published approaches for graph contrastive learning.

35th Conference on Neural Information Processing Systems (NeurIPS 2021).

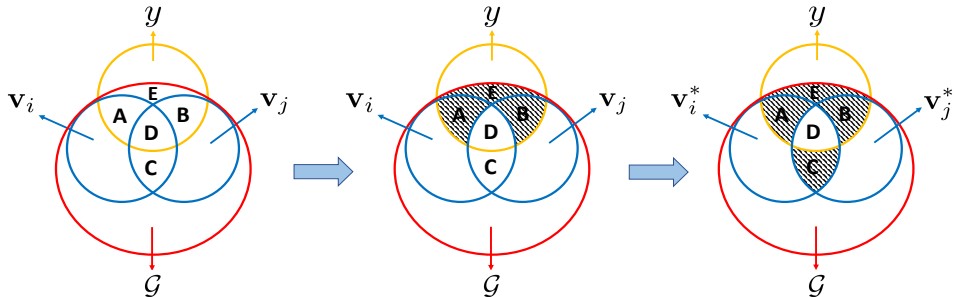

Figure 1: Illustration of optimal views.

## 2 Proofs of Corollaries

Basically, we follow the proofs in [5] to provide the proofs of our corollaries. The main differences are two-fold: 1) the input is graph $\mathcal{G}$ for our framework, and 2) the concept of optimal contrastive mode is unique for graph contrastive learning.

**Corollary 1.** *(Optimal Augmented Views) For a downstream task $T$ whose goal is to predict a semantic label $y$, the optimal views, $\mathbf{v}_i^*$, $\mathbf{v}_j^*$, generated from the input graph $\mathcal{G}$ are the solutions to the following optimization problem :*

$$(\mathbf{v}_i^*, \mathbf{v}_j^*) = \arg\min_{\mathbf{v}_i, \mathbf{v}_j} I(\mathbf{v}_i; \mathbf{v}_j) \tag{1}$$

$$s.t. \ \ I(\mathbf{v}_i; y) = I(\mathbf{v}_j; y) \tag{2}$$

$$I(\mathbf{v}_i; y) = I(\mathcal{G}; y) \tag{3}$$

*Proof.* Because Equation (3) holds and $v_i$, $v_j$ are functions of $\mathcal{G}$, it is natural to know $I(\mathcal{G}; y) = I(\mathbf{v}_i, \mathbf{v}_j; y)$ holds. Because $I(\mathbf{v}_i, \mathbf{v}_j; y) = I(\mathbf{v}_i; y) + I(\mathbf{v}_j; y|\mathbf{v}_i)$ and Equation (3) holds, we know $I(\mathbf{v}_i, \mathbf{v}_j; y) = I(\mathcal{G}; y) + I(\mathbf{v}_j; y|\mathbf{v}_i)$. Because $I(\mathcal{G}; y) = I(\mathbf{v}_i, \mathbf{v}_j; y)$ already holds and the nonnegativity of mutual information, we know

$$I(\mathbf{v}_j; y|\mathbf{v}_i) = 0. \tag{4}$$

Thus, we know $I(\mathbf{v}_i; \mathbf{v}_j) = I(\mathbf{v}_i; \mathbf{v}_j) + I(\mathbf{v}_j; y|\mathbf{v}_i) = I(\mathbf{v}_i, y; \mathbf{v}_j)$. In addition, we know $I(\mathbf{v}_i, y; \mathbf{v}_j) = I(\mathbf{v}_j; y) + I(\mathbf{v}_i; \mathbf{v}_j|y)$. Thus, $I(\mathbf{v}_i; \mathbf{v}_j) = I(\mathbf{v}_j; y) + I(\mathbf{v}_i; \mathbf{v}_j|y) \geqslant I(\mathbf{v}_j; y) = I(\mathcal{G}; y)$. Therefore, $I(\mathbf{v}_i; \mathbf{v}_j)$ can reach the minimum, which is $I(\mathcal{G}; y)$ And when $I(\mathbf{v}_i; \mathbf{v}_j)$ can reach the minimum, the optimal views are conditionally independent, which is described as

$$I(\mathbf{v}_i^*; \mathbf{v}_j^*|y) = 0. \tag{5}$$

$\square$

It is noticed that: 1) when Equation (4) holds, the area of B in Figure 1 becomes null. 2) when Equation (5) holds, the area of C in Figure 1 becomes null.

**Corollary 2.** *(Optimal View Encoder) Given the optimal views, $\mathbf{v}_i^*$, $\mathbf{v}_j^*$, for a downstream task $T$ whose goal is to predict a semantic label $y$, the optimal view encoder for view $\mathbf{v}_i^*$ is the solution to the following optimization problem :*

$$f_i^* = \arg\min_{f_i} I(f_i(\mathbf{v}_i^*); \mathbf{v}_i^*) \tag{6}$$

$$s.t. \ \ I(f_i(\mathbf{v}_i^*); \mathbf{v}_j^*) = I(\mathbf{v}_i^*; \mathbf{v}_j^*) \tag{7}$$

We can follow the basic idea of Proposition A.2 and Proposition A.3 in Appendix of [5] to get the proof of Corollary 2. The main difference is that we focus on the optimal view encoders, however, [5] focuses on the optimal representations.

**Corollary 3.** *(Optimal Contrastive Mode) Given the latent representations, $\mathbf{z}_i^*$, $\mathbf{z}_j^*$, extracted by the optimal view encoders, i.e., $\mathbf{z}_i^* = f_i^*(\mathbf{v}_i^*)$, $\mathbf{z}_j^* = f_j^*(\mathbf{v}_j^*)$ , and a downstream task $T$ with label $y$, the optimal contrastive mode is the solution to the following optimization problem, where $c_i$, $c_j$ are the aggregation operations applied to the latent representations:*

$$(c_i^*, c_j^*) = \arg\min_{(c_i, c_j)} -I(c_i(\mathbf{z}_i^*); c_j(\mathbf{z}_j^*)). \tag{8}$$

*Proof.* Recall the objective function of graph contrastive learning is to minimize the contrastive loss, e.g., $\mathcal{L}_{NCE}$. Minimizing the loss equivalently maximizes the mutual information between latent representations based on contrastive modes. For example, minimizing $\mathcal{L}_{NCE}$ equivalently maximizes $I(\mathbf{z}_i, \mathbf{z}_j)$, which is because $I(\mathbf{z}_i, \mathbf{z}_j) \geqslant log(N)$ - $\mathcal{L}_{NCE}$. Thus, we just need to iterate through all the modes and choose the mode with the largest mutual information value.

$\square$

# 3    Experimental Settings

The algorithm of InfoGCL for both graph and node classification is described in Algorithm 1. The view augmentation methods include node dropping, edge perturbation, attribute masking, subgraph sampling, and graph diffusion. The GNN backbones of view encoders include GCN, GAT, and GIN. The aggregation operations include identical transformation and taking average, which construct the global-global, local-global, local-local, multi-scale mode, and hybrid modes. It is noticed that the two views share the same view encoder (Proposition 2), which is because the domains of the two views are the same.

---

**Algorithm 1:** Training algorithm of InfoGCL for both graph and node classification

---

1: **Input:** The input graph $\mathcal{G}$, a task $T$ with label $y$, a bunch of graph view augmentation methods, $\{q_1(\cdot), q_2(\cdot), \cdots \}$, a set of view encoders, $\{f^1(\cdot), f^2(\cdot), \cdots \}$, a set of aggregation operations, $\{c_1(\cdot), c_2(\cdot), \cdots \}$.

2: **Output:** The recommended augmentation methods, $q_i^*(\cdot), q_j^*(\cdot)$, the recommended view encoder, $f^*(\cdot)$, the recommended contrastive mode, $(c_i^*(\cdot), c_j^*(\cdot))$.

3: **# Proposition 1: optimal view augmentations**

4: $L_a \leftarrow []$

5: **for** view augmentation $q_i(\cdot) \in \{q_1(\cdot), q_2(\cdot), \cdots \}$ **do**

6:     **for** view augmentation $q_j(\cdot) \in \{q_1(\cdot), q_2(\cdot), \cdots \}$ **do**

7:         $\mathbf{v}_i \leftarrow q_i(\mathcal{G})$

8:         $\mathbf{v}_j \leftarrow q_j(\mathcal{G})$

9:         $a \leftarrow I(\mathbf{v}_i; y) + I(\mathbf{v}_j; y) - I(\mathbf{v}_i; \mathbf{v}_j)$.

10:         Add a to $L_a$.

11:     **end for**

12: **end for**

13: Choose the augmentations with the maximum value in $L_a$ as $q_i^*(\cdot)$ and $q_j^*(\cdot)$, respectively.

14: **# Proposition 2: optimal view encoder**

15: $L_b \leftarrow []$

16: **for** view encoder $f(\cdot) \in \{f^1(\cdot), f^2(\cdot), \cdots \}$ **do**

17:     $\mathbf{z}_i \leftarrow f(\mathbf{v}_i)$

18:     $\mathbf{z}_j \leftarrow f(\mathbf{v}_j)$

19:     $b \leftarrow I(\mathbf{v}_i; \mathbf{z}_i; y) + I(\mathbf{v}_j; \mathbf{z}_j; y)$.

20:     Add b to $L_b$.

21: **end for**

22: Choose the encoder with the maximum value in $L_b$ as $f^*(\cdot)$.

23: **# Proposition 3: optimal contrastive mode**

24: $L_c \leftarrow []$

25: **for** aggregation operation $c_i(\cdot) \in \{c_1(\cdot), c_2(\cdot), \cdots \}$ **do**

26:     **for** aggregation operation $c_j(\cdot) \in \{c_1(\cdot), c_2(\cdot), \cdots \}$ **do**

27:         $c \leftarrow I(c_i(\mathbf{z}_i); c_j(\mathbf{z}_j); y)$.

28:         Add c to $L_c$.

29:     **end for**

30: **end for**

31: Choose the operations with the maximum value in $L_c$ as $c_i^*(\cdot)$ and $c_j^*(\cdot)$, respectively.

---

# 4 Quantitative Evaluation of Proposition

To test the validity of our proposed propositions, we conduct the ablation studies on the Cora and Citeseer datasets. Specifically, we use the GraphCL [7] as the backbone and test it on Proposition 1.

**Proposition 1.** *For a task $T$ with label $y$, given a bunch of graph view augmentation methods, $\{q_1(\cdot), q_2(\cdot), \cdots\}$, that create two views $\mathbf{v}_i$, $\mathbf{v}_j$, the recommended augmentation methods are the ones, $q_i(\cdot), q_j(\cdot)$, that maximize $I(\mathbf{v}_i; y) + I(\mathbf{v}_j; y) - I(\mathbf{v}_i; \mathbf{v}_j)$, i.e., the area of $A+B+D$ in Figure 1.*

| Augmented Views | Classification Accuracy | | $I(\mathbf{v}_i; y) + I(\mathbf{v}_j; y) - I(\mathbf{v}_i; \mathbf{v}_j)$ | |
|---|---|---|---|---|
| | Cora | Citeseer | Cora | Citeseer |
| EdgePert vs. Identical | $82.5 \pm 0.1$ | $72.2 \pm 0.2$ | $4.5671 \pm 0.1$ | $812.5062 \pm 0.6$ |
| EdgePert vs. EdgePert | $82.3 \pm 0.2$ | $73.1 \pm 0.2$ | $4.1921 \pm 0.4$ | $821.3120 \pm 1.0$ |

Table 3: The comparison between classification accuracy and $I(\mathbf{v}_i; y) + I(\mathbf{v}_j; y) - I(\mathbf{v}_i; \mathbf{v}_j)$ for different augmented views. $I(\mathbf{v}_i; y) + I(\mathbf{v}_j; y) - I(\mathbf{v}_i; \mathbf{v}_j)$ is approximated by $- (\text{CE}_i + \text{CE}_j - (\text{CE}'_i + \text{CE}'_j))$.

To calculate $I(\mathbf{v}_i; y) + I(\mathbf{v}_j; y) - I(\mathbf{v}_i; \mathbf{v}_j)$, we use cross-entropy to approximate the mutual information. Specifically, we feed $\mathbf{v}_i$ into GNNs and the outputs are further fed into a MLP to generate logits. We calculate the cross-entropy, $\text{CE}_i$, based on the logits and $y$. Similarly, we are able to calculate the cross-entropy, $\text{CE}_j$, based on $\mathbf{v}_j$ and $y$. To approximate $I(\mathbf{v}_i; \mathbf{v}_j)$, we first feed $\mathbf{v}_i$, $\mathbf{v}_j$ into another GNNs and MLP. Then, we use the outputs of $\mathbf{v}_i$ as logits and the outputs of $\mathbf{v}_j$ as labels to get $\text{CE}'_i$. Symmetrically, we are able to get $\text{CE}'_j$. Thus, we use $- (\text{CE}_i + \text{CE}_j - (\text{CE}'_i + \text{CE}'_j))$ to approximate $I(\mathbf{v}_i; y) + I(\mathbf{v}_j; y) - I(\mathbf{v}_i; \mathbf{v}_j)$. We use two-layer GCNs as the GNNs and $\text{CE}_i + \text{CE}_j - (\text{CE}'_i + \text{CE}'_j)$ as the loss function. We adopt the default settings and train the model until it converges. The results are reported in Table 3. It is observed that, on the Cora dataset, the view pair of edge perturbation and the graph itself shows a little higher performance than the pair of edge perturbation and edge perturbation. According to the right part of Table 3, we see the view pair of edge perturbation and the graph itself has a greater value of $I(\mathbf{v}_i; y) + I(\mathbf{v}_j; y) - I(\mathbf{v}_i; \mathbf{v}_j)$ than the pair of edge perturbation and edge perturbation. Similar observations can be observed on the Citeseer dataset. Thus, the validity of Proposition 1 is verified.

**Proposition 2.** *Given a task $T$ with label $y$ and a set of view encoders, $\{f_i^1(\cdot), f_i^2(\cdot), \cdots\}$, that generate representation $\mathbf{z}_i$ via taking view $\mathbf{v}_i$ as input, the recommended view encoder is the one that maximizes the mutual information between $\mathbf{v}_i$, $\mathbf{z}_i$ and $y$. Symmetrically the same for view $\mathbf{v}_j$.*

| GNN Backbone | Classification Accuracy | | Mutual Information | |
|---|---|---|---|---|
| | Cora | Citeseer | Cora | Citeseer |
| GCN | $82.5 \pm 0.1$ | $72.2 \pm 0.2$ | $30.8579 \pm 1.1$ | $52.6575 \pm 0.9$ |
| GAT | $83.1 \pm 0.4$ | $73.1 \pm 0.5$ | $30.9037 \pm 0.5$ | $56.0916 \pm 0.6$ |

Table 4: The comparison between classification accuracy and the mutual information ($I(\mathbf{v}_i; \mathbf{z}_i; y) + I(\mathbf{v}_j; \mathbf{z}_j; y)$) for different GNN backbones. $I(\mathbf{v}_i; \mathbf{z}_i; y)$ is approximated by $I(\mathbf{v}_i; \mathbf{z}_i) + I(\mathbf{v}_i; y) + I(\mathbf{z}_i; y)$. Similarly for $I(\mathbf{v}_j; \mathbf{z}_j; y)$.

We follow a similar way to approximate the mutual information. Specifically, $I(\mathbf{v}_i; \mathbf{z}_i; y)$ is approximated by $I(\mathbf{v}_i; \mathbf{z}_i) + I(\mathbf{v}_i; y) + I(\mathbf{z}_i; y)$ and $I(\mathbf{v}_j; \mathbf{z}_j; y)$ is approximated by $I(\mathbf{v}_j; \mathbf{z}_j) + I(\mathbf{v}_j; y) + I(\mathbf{z}_j; y)$, which is because these views are approximate optimal views generated by Proposition 1.