# OpenReview forum: "InfoGCL: Information-Aware Graph Contrastive Learning"
_NeurIPS.cc/2021/Conference — NeurIPS 2021 Poster_

### Official Review · Reviewer_gMsc · 2021-07-17

**Rating:** 7
**Confidence:** 4

**Summary:**

In this paper, the authors study the graph contrastive learning problem and propose an information-aware approach called InfoGCL. The motivation for this work is that all recent graph contrastive learning approaches are customized, and it is an open question how to build the graph contrastive learning method for particular tasks and datasets. The authors' method decouples the typical graph contrastive learning into three modules and formalizes how to find the optimal of the three modules into three optimization problems. Experimental results on both node-level and graph-level tasks demonstrate the advantages of the proposed method.

**Limitations And Societal Impact:**

Yes

**Main Review:**

This paper is pretty interesting overall. Different from existing work, this work tries to unify the various contrastive learning approaches for graph-structure data. It extends the information bottleneck theory to optimize the representation learning in the graph contrastive learning framework.

One main contribution of this work is to understand graph contrastive learning from the optimization view. From the observations in section 5.3, it seems that the proposed InfoGCL is general and well supported by existing work. Personally, I like this work because instead of simply proposing a method to achieve better results, it digs into the hidden reasoning behind graph contrastive learning.

This paper contains solid theoretical content, which is well organized and easy to follow. The motivation, contribution, and methodology are well written. The experimental results support the claims. Overall, this paper is with good writing and clarity.

However, there are some specific questions/concerns.

1. In the Introduction, “Second, graph data has unique properties that distinguish it from other types of data, such as rich structural information and highly diverse distribution”. I see the proposed method considers graph structure, but not clear how it could generalize to cover various graph properties.

2. In the Introduction, “However, different from the typical representation learning, there are two information flows involved in the two augmented views in contrastive learning.”. I am not an expert in the information theory area. Could the authors elaborate more on the difference/significance in terms of the two flows?

3.  Corollary 1 (Section 4.1). Are you sure to achieve optimal conditions in real data scenarios? It looks very constrained.

4. In Section 4.3, “The main motivation is: the optimal contrastive model keeps the most task-relevant information after the representations are aggregated”. This problem appears to be a discrete optimization problem, and the authors have used the method of obtaining the optimum by traversal. Can there be a more optimal solution here?

5. In Section 5.4, “Therefore, we make the hypothesis: negative samples benefit graph modeling, especially when i) network topology, and ii) node features are extremely sparse.”. I am interested in the study of negative samples. I have not seen these conclusions in the cv domain. Can the authors provide more analytical or experimental supports?

6. In Section 5 (Experiments). The datasets of Cora Citeseer Pubmed are very small. I suggest the authors experiment with larger datasets.

7. The propositions in Section 4.4 are interesting and seem reasonable. More evidence/support is preferred.



**Time Spent Reviewing:**

3

---

> ### Author Response · Authors · 2021-08-10
> **Response to reviewer gMsc**
>
> Thank you for the valuable comments. Please find our responses as follows.
>
> Q1: How does the proposed method generalize to cover other graph properties?\
> A1: Thanks for raising this point. Our method is flexible and can handle various graph properties via adjusting the modules of view augmentation and view encoding: 1) We follow [R1] to apply four augmentation approaches and different approaches focusing on different graph properties. For example, attribute masking covers node attribute information and subgraph sampling focuses on local topology information. 2) Adapting different kinds of view encoders can further help handle different graph properties effectively [R2]. Besides, experimental results on various graph datasets verify the ability of our method to cover different graph properties.
>
> [R1] You, Y., Chen, T., Sui, Y., Chen, T., Wang, Z., & Shen, Y. (2020). Graph contrastive learning with augmentations. NeurIPS.\
> [R2] Zhang, Z., Cui, P., & Zhu, W. (2020). Deep learning on graphs: A survey. IEEE TKDE.
>
> Q2: What is the difference of information flows between contrastive learning and typical representation learning?\
> A2: This is an interesting study part. For typical representation learning, there is only one information flow. For example, the input data is fed into an encoder and then the generated representation is fed into a downstream task operator. However, there are two information flows in contrastive learning for two augmented views respectively. The two flows interact with each other at the representation contrasting stage [R3]. In addition, the graph contrastive learning is different from other types of data, such as rich structural information and highly diverse distribution [R4, R5]. Thus, it is desirable and challenging to design the contrastive learning model for various graph datasets, which motivates this work.
>
> [R3] Tian, Y., Sun, C., Poole, B., Krishnan, D., Schmid, C., & Isola, P. (2020). What makes for good views for contrastive learning?. NeurIPS.\
> [R4] Veličković, P., Fedus, W., Hamilton, W. L., Liò, P., Bengio, Y., & Hjelm, R. D. (2019). Deep graph infomax. ICLR.\
> [R5] Sun, F. Y., Hoffmann, J., Verma, V., & Tang, J. (2019). Infograph: Unsupervised and semi-supervised graph-level representation learning via mutual information maximization. ICLR.
>
> Q3: How about the optimal conditions for Corollary 1 (Optimal Augmented Views) in real data scenarios?\
> A3: Thanks for raising this point. Yes, the exactly optimal conditions for getting the optimal views are indeed hard to meet. As shown in Figure 2, to reach the optimal views, there are two steps. First, as shown in the middle of Figure 2, the two views should share the same amount of task-relevant information. Second, as shown in the right of Figure 2, all the shared information between views is task-relevant. The proof is in the Appendix. In real-world scenarios, the generated views will contain a lot of noise inside, which makes them hard to be optimal. However, under our setting of view augmentation, we can control the amount of shared information between views. For example, following [R1], we can adjust the amount of information shared between the two views by controlling the edge dropping ratio. Therefore, we can approximate the optimal views in practice.
>
> Q4: Is there a better solution to get the optimal contrastive mode?\
> A4: Yes, there could be a better solution if the aggregation operations can be represented by learnable parameters (e.g., neural networks) and the optimal contrastive model would be achieved via the typical optimization method (e.g., gradient descent). However, under our current setting, we only focus on a batch of given operations, i.e., the four modes in Section 4.3. Therefore, it is easier and more practical to get the optimal contrastive mode by traversal. We will extend the setting to make it more general in our future work.
>
> Q5: Can the authors provide more analytical or experimental supports for the influence of negative samples?\
> A5: Yes. The main reason for the different performance of our method on the three graph task datasets and the three node-task datasets used in our paper is about the sparsity of graph data.  The three node-task datasets are very sparse in terms of both graph topology and node features (one-hot encoding with high dimensionality). Therefore, if only positive samples are utilized, the learning models would tend to collapse because it is difficult to learn effective representations from such sparse data, and therefore the representations cannot be used to distinguish the properties of different nodes or graphs. However, for the graph datasets used in our work, the distributions of node features and graph topology are dense. Even without negative samples, the properties of different nodes or graphs can be captured, which is similar to the observations in SimSiam [R6].
>
> [R6] Chen, X., & He, K. (2021). Exploring simple siamese representation learning. In Proceedings of the IEEE/CVF Conference on CVPR.
>
> Q6: The datasets of Cora, Citeseer, Pubmed are very small.\
> A6: Thanks for raising this point. Our main reason for using the three datasets is that they are used by many existing methods and facilitate our experimental comparisons. We will add more experiments on larger node task datasets.
>
> Q7: More evidence/support for the propositions in Section 4.4 is preferred.\
> A7: We provide both qualitative and quantitative evaluations of these propositions in the experimental part (Section 5.3) and Appendix (Section 4). We will add more empirical studies to verify these propositions.

---

### Official Review · Reviewer_U1jZ · 2021-07-20

**Rating:** 7
**Confidence:** 4

**Summary:**

This paper proposes InfoGCL, a general framework for graph contrastive learning framework design. More specifically, InfoGCL investigates the optimal (and practically optimal) option for three modules in graph contrastive learning, i.e. view augmentation, view encoding and representation contrasting. Experimental results have proven effective over various benchmarks compared with groups of the kernel, supervised and unsupervised.

**Main Review:**

Strengths:
1. Good formulation and clear identification of the three-stage graph contrastive learning process.
2. Technically sound through the angle of information bottleneck of optimal graph contrastive learning options, with improved explainability.
3. Extensive experimental on InfoGCN over a large group of baseline through fair comparison with important observations.

Weaknesses:
1. A few terms and concepts that may not be generally recognized are not formally defined or fully discussed.
2. IB-based InfoGL seems to be only applicable for a single (given) task.
3. Some experimental results are less convincing and somewhat lack sufficient explanation.

Detailed comments and questions:
1. As pointed out in the weaknesses, it is assumed that all the proposition 1-3 are applied when only one task T is pre-given, while many GNN frameworks are intended for multiple downstream tasks while graph contrastive learning is served for pre-training. One drawback is that, once the views are optimized over “task-related” information, the InfoGCL would have to perform independently (or train from scratch) for multiple downstream tasks.
2. Regarding experiment:
2.1  The variances of Node2vec and sub2vec on the MUTAG dataset are extremely high which seems strange.
2.2 The performance drop without negative samples on the Cora dataset and others is insufficient and not convincing (even though the authors only hypothesize the sparsity is one factor).
3. Many terms that are not universally recognized or self-explained are not formally defined and may need clarification. Examples are:
3.1 The term “view” i.e. generated variations (subgraphs) from the original graph which may have confusing meanings in other graph communities.
3.2 The Venn figures in Figures 1 and 2 need revision for better visibility and explanation.
3.3 The detailed definition of 5 modes in Section 4.3
3.4 “Negative pairs” in InfoNCE loss vs “Negative samples” in Section 4.5

Minor corrections or suggestions:
1. The Algorithm 1 block in the appendix is relatively important to understand the practical implementation of InfoGCL, which is recommended to be added in the main section. Otherwise, reviewers may be lost about how InfoGCL is working in practice.
2. Line 172: Typo “gprah”



**Time Spent Reviewing:**

1.5

---

> ### Author Response · Authors · 2021-08-10
> **Response to reviewer U1jZ**
>
> Thank you for the thorough review. Please see our responses below:
>
> Q1: One drawback is that the proposed method has to perform independently (or train from scratch) for multiple downstream tasks. \
> A1: This is an interesting study part. Yes, this is one drawback that the proposed method is task-specific. This property is caused by our adoption of the Information Bottleneck principle, which requires that the downstream task is given. For example, both [R1] and [R2] apply Information Bottleneck to the graph domain (not contrastive learning). They aim to learn graph representations for a given task or label. [R3] applies Information Bottleneck to the computer vision domain and also requires that the task is given in advance.
>
> However, this drawback of our proposed method could be addressed. One potential solution is to train the same view encoder networks based upon the contrastive losses of different downstream tasks, which facilitates the generalization ability of the encoder. Thus, the encoder could be further used to generate representations for different tasks. The other solution is that we can use the well-trained encoder networks for one task as the initialized encoder for another task if the two tasks have a certain degree of similarity, e.g., both tasks focus on learning the node-level representation. We will explore how to improve the generalization ability and robustness of the proposed method in the future work.
>
> [R1] Wu, T., Ren, H., Li, P., & Leskovec, J. (2020). Graph information bottleneck. NeurIPS. \
> [R2] Yu, J., Xu, T., Rong, Y., Bian, Y., Huang, J., & He, R. (2021). Graph information bottleneck for subgraph recognition. ICLR. \
> [R3] Tian, Y., Sun, C., Poole, B., Krishnan, D., Schmid, C., & Isola, P. (2020). What makes for good views for contrastive learning?. NeurIPS.
>
> Q2: The variances of node2vec and sub2vec on the MUTAG dataset. \
> A2: Thanks for raising this point. We exactly use the same experimental results of node2vec and sub2vec as reported in [R4, R5]. However, we believe exploring how to improve stability is an important research topic. There is also some existing related work on this topic [R6, R7], and we will study how to reduce the variance of the baseline methods to make a fairer comparison.
>
> [R4] Hassani, K., & Khasahmadi, A. H. (2020). Contrastive multi-view representation learning on graphs. ICML. \
> [R5] You, Y., Chen, T., Sui, Y., Chen, T., Wang, Z., & Shen, Y. (2020). Graph contrastive learning with augmentations. NeurIPS.\
> [R6] Chen, J., Zhu, J., & Song, L. (2018). Stochastic training of graph convolutional networks with variance reduction. ICML.\
> [R7] Ma, K., Yang, H., Yang, H., Jin, T., Chen, P., Chen, Y., ... & Cheng, J. (2021). Improving graph representation learning by contrastive regularization. arXiv preprint arXiv:2101.11525.
>
> Q3: The performance drop without negative samples on Cora and other datasets is insufficient and not convincing.\
> A3: Thanks for raising this point. We admit that the performance drop without negative samples on Cora, Citeseer, Pubmed might not be convincing enough. Because there may be other factors that contribute to the performance drop. However, according to our experimental results (Table 4), the performance drops on Cora, Citeseer, Pubmed are -8.2%, -3.1%, -3.0%, respectively. In contrast, the drops on three graph-task datasets (MUTAG, IMDB-B, COLLAB) are -0.2%, 0.0%, +0.2%, respectively. Thus, we think the conclusion that performance drop is sufficient holds, according to the experimental setup in our work. We will add more experiments to verify this conclusion.
>
> Q4: The term “view” may have confusing meanings in other graph communities.\
> A4: Thanks for raising this point. We will revise the term used to describe the generated graph variations and define them more formally.
>
> Q5: The Venn figures in Figures 1 and 2 need revision.\
> A5: Thanks for raising this point. We will revise Figures 1-2 and add more explanations to make them easier to follow.
>
> Q6: The detailed definition of 5 modes in Section 4.3 (Representation Contrasting).\
> A6: Thanks for raising this point. We will add detailed definitions of the five modes. Basically, we follow [R4] to design the five modes (global-global, local-global, local-local, multi-scale, hybrid). “global” represents the graph-level representation, “local” represents the node-level representation, “multi-scale” indicates that we contrast graph representation of one view with the representation of intermediate graph structure from the other view, and “hybrid” indicates both global-global and local-global are applied.
>
> Q7: “Negative pairs” in InfoNCE loss vs “Negative samples” in Section 4.5.\
> A7: Thanks for raising this point. A negative pair is a pair of representations. Typically, one representation is from an augmented view of a selected graph. The other is constructed from the augmented views of other graphs in the same minibatch. In contrast, the negative samples represent all the representations that are constructed from other graphs in the same minibatch. We will add formal definitions for them.
>
> Q8: The Algorithm 1 block in the appendix is recommended to be added in the main section.\
> A8: Thanks for the suggestion. We will revise Algorithm 1 and add it in the main section if there is enough space. In addition, we will add more descriptions in the main section about the pipeline of our proposed method to help people understand how our method works in practice.
>
> Q9: Typos, including “gprah”.\
> A9: Thanks for raising this point. We will proofread our work more and address these typos, such as “gprah” in Line 172, “tends” in Line 313, etc.

---

> > ### Comment · Reviewer_U1jZ · 2021-09-10
> > **I have read the response.**
> >
> > I have read the author's replies for my review concerns as well as other reviews and discussions. The authors generally addressed my questions and provided a satisfactory response to my comment. I would stick to my current score evaluation considering the innovation, significance, and clarity.

---

### Official Review · Reviewer_PEad · 2021-07-24

**Rating:** 7
**Confidence:** 2

**Summary:**

The paper provides a systematic approach to building graph contrastive learning models for
any particular graph learning taks and dataset. The suggested approach builds on the Information Bottleneck principle and unifies all recent graph representation learning methods based on contrastive learning.


**Limitations And Societal Impact:**

Limitations are mentioned in the final section, but not explicitly explained.
Societal Impact is missing.

**Main Review:**

Overall I think this is a good paper, trying to present a unifying view on graph contrastive learning. A typical graph contrastive learning model is decoupled into three sequential modules: View augmentation, view encoding and representation contrasting. Based on the principle of the Information Bottleneck, the paper suggest feasible objectives to identify optimal modules for each of these three parts:
* The augmented views should contain as much task-relevant information as possible, while they shoul share as little information as possible.
* The view encoder should be task-relevant and simple as much as possible.
* Thhe contrastive mode should keep task-relevant information as much as possible70after contrasting

The paper also argues that negative samples are not necessarily required for graph contrastive learning.

Originality: The paper unifies existing approaches, and provides a novel systemic perspective on graph contrastive learning.

Quality: The paper is technically sound, corollaries are proven in the the appendix, the work is complete as such.

Clarity: The paper is written well.

Significance: The results are important to researches in the area of graph representation learning. It's holistic approach to graph contrastive learning through the lens of the Information Bottleneck principle provides novel insights for this area of research. The presented empirical results look convincing.

A minor detail is Eq.9: I think it would be nice to explain with a few sentences what the principle behind SimSiam is.

**Time Spent Reviewing:**

2.5h

---

> ### Author Response · Authors · 2021-08-10
> **Respond to reviewer PEad**
>
> We thank the reviewer for acknowledging the value of our work. In the following, please find our response to the question.
>
> Q1: Explain what the principle behind SimSiam [R1] is. \
> A1: Thanks for raising this point. Overall, SimSiam is a kind of simple Siamese networks [R2] that can effectively learn unsupervised visual representations. SimSiam is a self-supervised learning approach.
>
> The problem with simple self-supervised representation learning is that after a few epochs, the model collapses to identity mapping because the loss would be zero. There are usually three ways to solve this problem: using negative samples [R3], applying online clustering [R4], and applying moving average [R5]. However, SimSiam can work surprisingly well with none of the above strategies for preventing model collapsing. The main technique SimSiam adopts is the stop-gradient operation, which only updates one branch at the training step. It is shown that the stop-gradient operation is sufficient to avoid collapsing solutions.
>
> However, current graph contrastive learning approaches are highly dependent on negative samples. Motivated by SimSiam and [R3] which indicate that negative samples are not necessarily required in the vision domain, of which the main benefit is to avoid careful treatment to retrieve the negative pairs, we explore the necessity of negative samples in the graph domain. Therefore, we follow the framework of SimSiam to revise the loss function as Eq. (9). Our experimental results indicate that negative samples are not necessarily required for graph contrastive learning. But negative samples would still work in some cases, such as when the graph data is very sparse in terms of graph topology and node features.
>
> [R1] Chen, X., & He, K. (2021). Exploring simple siamese representation learning. In Proceedings of the IEEE/CVF Conference on CVPR. \
> [R2] Bromley, J., Guyon, I., Lecun, Y., Säckinger, E., & Shah, R. (1994). Signature verification using a “Siamese” time delay neural network. NeurIPS. \
> [R3] Chen, T., Kornblith, S., Norouzi, M., & Hinton, G. (2020). A simple framework for contrastive learning of visual representations. ICML. \
> [R4] Caron, M., Misra, I., Mairal, J., Goyal, P., Bojanowski, P., & Joulin, A. (2020). Unsupervised learning of visual features by contrasting cluster assignments. NeurIPS. \
> [R5] Grill, J. B., Strub, F., Altché, F., Tallec, C., Richemond, P. H., Buchatskaya, E., ... & Valko, M. (2020). Bootstrap your own latent: A new approach to self-supervised learning. NeurIPS.

---

### Public Comment · ~Mert_Kosan1 · 2022-06-24
**Reproducibility - Code Sharing is Needed.**

Hello,

For reproducibility and running the method on different datasets, can you share the paper's code?

Best,
Mert

---

### Decision · Program_Chairs · 2021-09-27

**Decision:**

Accept (Poster)

**Comment:**

The paper proposes a systematic approach to contrastive learning on graphs which builds on ideas from information bottleneck theory and unifies multiple graph representation learning methods in a single framework. The paper is written well and relevant to the NeurIPS community. All reviewers support its acceptance, especially due to the novel and promising approach which unifies existing methods and provides new perspectives for graph learning. Reviewers highlighted also that the approach is technically sound, that claims in the paper are theoretically supported, and convincing experimental results. Please take the feedback on concerns and limitations from reviewers into account when preparing the camera ready version of the manuscript.